# Unravelling the changes during induced vitellogenesis in female European eel through RNA-Seq: What happens to the liver?

Francesca Bertolini[1]*, Michelle Grace Pinto Jørgensen[1], Christiaan Henkel[2], Sylvie Dufour[3], Jonna Tomkiewicz[1]

1 National Institute of Aquatic Resources, Technical University of Denmark, Lyngby, Denmark, 2 Faculty of Veterinary Medicine, Norwegian University of Life Sciences, Oslo, Norway, 3 Laboratory BOREA, Museum National d'Histoire Naturelle, CNRS, Sorbonne University, Paris, France

* franb@aqua.dtu.dk

**Data Availability Statement:** Raw RNA-Seq reads can be found in ENA (European Nucleotide Archive) under accession number PRJEB37006.

## Abstract

The life cycle of European eel (*Anguilla anguilla*), a catadromous species, is complex and enigmatic. In nature, during the silvering process prior to their long spawning migration, reproductive development is arrested, and they cease feeding. In studies of reproduction using hormonal induction, eels are equivalently not feed. Therefore, in female eels that undergo vitellogenesis, the liver plays different, essential roles being involved both in vitellogenins synthesis and in reallocating resources for the maintenance of vital functions, performing the transoceanic reproductive migration and completing reproductive development. The present work aimed at unravelling the major transcriptomic changes that occur in the liver during induced vitellogenesis in female eels. mRNA-Seq data from 16 animals (eight before induced vitellogenesis and eight after nine weeks of hormonal treatment) were generated and differential expression analysis was performed comparing the two groups. This analysis detected 1,328 upregulated and 1,490 downregulated transcripts. Overrepresentation analysis of the upregulated genes included biological processes related to biosynthesis, response to estrogens, mitochondrial activity and localization, while downregulated genes were enriched in processes related to morphogenesis and development of several organs and tissues, including liver and immune system. Among key genes, the upregulated ones included vitellogenin genes (*VTG1* and *VTG2*) that are central in vitellogenesis, together with *ESR1* and two novel genes not previously investigated in European eel (*LMAN1* and *NUPR1*), which have been linked with reproduction in other species. Moreover, several upregulated genes, such as *CYC1*, *ELOVL5*, *KARS* and *ACSS1*, are involved in the management of the effect of fasting and *NOTCH*, *VEGFA* and *NCOR* are linked with development, autophagy and liver maintenance in other species. These results increase the understanding of the molecular changes that occur in the liver during vitellogenesis in this complex and distinctive fish species, bringing new insights on European eel reproduction and broodstock management.

**Funding:** This work was supported by the Innovation Fund Denmark [grant numbers 5184-00093B (EEL-HATCH) and 7076-00125B (ITS-EEL)] to JT.

**Competing interests:** The authors have declared that no competing interests exist.

## Introduction

The life cycle of European eel (*Anguilla anguilla*), a catadromous species, is complex and spans a wide range of geographical areas with highly diverse habitats. European eels spend several life stages from glass eel to yellow eel, the so-called continental-phase, in freshwater and coastal areas of Europe and northern Africa, for a period ranging from 5 to 20 years [1]. After this period, a process called silvering prepares the future spawners (silver eels) for their oceanic reproductive migration that will lead them to their spawning areas in the Sargasso Sea [2,3]. However, silver eels are still sexually immature when they leave the continental habitats and their development remains blocked at this prepubertal stage as long as they remain in continental habitats [4,5]. In fact, both a low stimulation by gonadotropin-releasing hormone neurons and a strong dopaminergic inhibition maintain the production of pituitary gonadotropins at a low level, thus preventing the eels to complete the sexual maturation process presumably until they reach the Sargasso Sea [6]. During silvering, several phenotypical and physiological changes occur: for example, the abdominal skin color changes from yellow to silver, the pectoral fins darken and the eyes increase in size [7]. At this stage, they stop feeding, the digestive tract regresses, while several metabolic changes occur [8–10]. Consequently, European eels need to reallocate and use accumulated reserves for the migration to their spawning areas and their reproductive development.

Since the 1980's, the European eel stock has drastically declined due to reduction of habitats, obstacles to migrations including hydropower plants, fisheries, pollutants and possibly climate change [11,12]. Consequently, the species is now ranked as critically endangered [13], trade restricted [14] and an EU-wide management strategy is being implemented [15]. Thus, while landings and aquaculture production were high by the end of 2000's, eel has turned into a high value niche product in several European countries (Netherlands, Italy, and Denmark among others; [16]). Together, this has led to increased efforts in closing the life cycle of European eel in captivity, in the attempt to relieve the pressure from fisheries and to develop a sustainable aquaculture, which is also happening to the closely related anguillid species, the Japanese eel *Anguilla japonica* [17]. New advancements in induced sexual maturation have led to the development of effective assisted reproduction strategies using hormonal treatments to induce gamete development [18–20] in European eel, enabling production of viable embryos and larvae entering the feeding larval stage [21,22]. In the attempt to mimic the natural conditions during reproduction experiments, as a common and consolidated practice eels are not fed.

Whether induced in captivity or occurring naturally in their native habitat, vitellogenesis in female European eel, like other teleosts is regulated by the hypothalamus-pituitary-gonad-liver axis, where the liver plays several key roles. One of the liver's major roles is to provide key substances required by the developing oocyte, particularly vitellogenins that are precursors of yolk proteins. The biosynthesis of vitellogenins is mainly controlled by the gonadotropin follicle-stimulating hormone (FSH) that stimulates the synthesis by ovarian follicle cells of 17β-estradiol (E2), which in turn stimulates the production of vitellogenins through the interaction of E2 with its receptors in the liver (reviewed by [23]). Vitellogenins are then transported to the ovary through the bloodstream and accumulates in the developing oocyte as yolk granules or globules [24]. Moreover, as the eels do not feed, the liver is involved in the reallocation of resources such as lipids, energy metabolism and glycogen production for survival, trans-oceanic migratory swimming activity as well as completing reproductive development [25].

Transcriptomic changes in the liver during reproductive development remains little investigated in European eel. So far, one study that targeted male spermatogenesis and more than 14,000 transcripts, detected expression changes of several genes involved in key biological

pathways, e.g. lipid metabolism, fatty acid synthesis and transport, mitochondrial function, steroid transport and bile acid metabolism [26]. The technological advancements and the availability of the first genome assembly and annotation [27] provide the opportunity to increase the number of such types of high-throughput analysis, such as RNA-Seq, in this species. This has allowed to conduce broader gene expression investigations that has lead, for example, to the first characterizations and differentiations of the pituitary and ovary gland among yellow and silver eels [28,29].

In this context, the aim of this work was to unravel the transcriptomic changes that occur in the liver of female European eels during induced vitellogenesis, comparing female eels in the immature stage with hormonally treated eels with developed ovaries characterized by oocytes late vitellogenic stage.

## Materials and methods

### Eel maturation and sample collection

All experimental protocols were approved by the Animal Experiments Inspectorate (AEI), Danish Ministry of Food, Agriculture and Fisheries (permit number: 2015-15-0201-00696) and fish were handled according to the European Union regulations regarding protection of experimental animals (Dir 86/609/EEC).

Farmed female eels (n = 38, total length 76 ± 136 cm, body weight 900 ± 102 g) were reared in freshwater at a commercial eel farm (Stensgård Eel Farm A/S, Randbøl, Denmark) and were transported in an aerated freshwater tank to the research facility EEL-HATCH, Technical University of Denmark. The eels were placed in three tanks of 1,150 L connected to a recirculating aquaculture system. The eels were acclimatized to saltwater (i.e., from 0 to 36 psu) over two weeks by adding natural seawater supplemented with Blue Treasure Aquaculture Salt (Qingdao Sea-Salt Aquarium Technology Co., Ltd., Quindao, China). Water temperature was maintained at ~20˚C and light in a 12 h light and 12 h dark light regime with low light intensity (~20 lux W) throughout the experiment.

After two weeks acclimation, eight female eels (total length 75 ± 4 cm, body weight 862 ± 100 g) were randomly selected for sampling prior hormonal treatment (i.e., week 0). The remaining eels were anesthetized individually in an aqueous solution of benzocaine (ethyl p-aminobenzoate, 20 mg L$^{-1}$, Sigma Aldrich) and tagged with a passive integrated transponder (PIT, 12×2 mm) tag in the dorsal musculature for ID. Vitellogenesis was induced in these eels through weekly intramuscular injections with a constant dose (20 mg/kg initial body weight) of carp pituitary extract (CPE; Ducamar Spain S.L.U.). One week after the 9[th] injection (i.e., week 9), eight females (total length 75 ± 2 cm, body weight 901 ± 62 g) were randomly selected for sampling. All selected eels were euthanized by submergence in an aqueous solution of benzocaine (20 mg L$^{-1}$) for 5–10 min.

From each of the 16 sacrificed eels, hepatic tissue samples were collected and stored in RNA later at 4˚C for 24 h and then –20˚C until extraction. In addition, ovarian tissue for histological analyses was preserved in 4% sodium-buffered formalin (Hounisen, Skanderborg, Denmark) and stored at room temperature. Moreover, phenotypic parameters such as body weight, gonad and liver weight were recorded, and the hepatosomatic index (HSI) and gonadosomatic index (GSI) [30] were calculated as follows:

$$HSI = \text{liver weight/body weight}*100$$

$$GSI = \text{gonad weight/body weight}*100$$

Correlation among HSI and GSI was calculated with the ggpubr R package (https://CRAN.R-project.org/package=ggpubr).

## Histological analysis for determination of oocyte and ovarian developmental stage

Fixed tissues were dehydrated and embedded in paraffin using a sectioned at 5 μm using standard procedures. Sections were mounted on glass slides, stained with a 0.5% periodic acid solution, Schiff's reagent, Weigert's hematoxylin and metanil yellow [31]. Progression of ovarian development was categorized based on cytological characteristics of the most advanced cohort of oocytes, distinguishing previtellogenic and vitellogenic stages [32].

## RNA-Seq data generation, processing, alignment and read count

RNA was extracted from hepatic tissue with NucleoSpin® RNA (Macherey-Nagel, Germany) according to the manufacturer's instruction. RNA purity and quantity (260/280 = 2.17 ± 0.04, 260/230 = 2.26 ± 0.12) were evaluated through spectrophotometry using Nanodrop (Thermo Fisher Scientific, USA). Then, 2 μg of each RNA sample was sent to Novogene company (Beijing, China) to check for RNA integrity (RIN) through Bioanalyzer (Agilent Technologies, Santa Clara, CA), that assessed RIN > 8.2 for all samples. Paired end 150bp mRNA sequencing was then performed by the same company using Illumina HiSeq 2500 platform following the manufacturer's instruction (Illumina Inc, USA).

Quality of the generated reads was assessed through FastQC 0.11.4 [33]. The company pretrimmed reads from the adapters, but a further trimming was performed with Trimmomatic 0.38 [34] with the following parameters: HEADCROP:9 MINLEN:36 SLIDINGWINDOW:4:15. Only mate-paired trimmed reads were considered for downstream analyses. Trimmed reads were aligned to the European eel draft reference genome [27] using Tophat 2.0.13 [35] and including the GFF file with known transcripts available at https://doi.org/10.18710/L7GO8T. Uniquely mapped reads were then retained using Samtools 1.2 [36]. HTSeq-0.6.1 [37] was used to count reads for each gene with parameters "-m intersection-strict—stranded = no".

## Differential expression and over representation

Differential expression among week 0 and week 9 was analysed with the R Package Deseq2 [38]. Principal Component Analysis (PCA) was calculated based on the regularized log-transformation of read count with the same software. Differentially expressed genes (DEGs) with Padj < 0.01 was chosen as a threshold, according to what has been previously utilized by Burgerhout et al. [29] in a differential expression analysis of ovary tissues of the same species. Retained genes were then filtered as the status of the assembled genome is not complete and some genes may be redundant. Therefore, transcripts with the same gene symbol that showed opposite sign of Log2 fold change were removed.

Statistical overrepresentation test was performed with upregulated and downregulated genes separately using Panther (http://www.pantherdb.org/), considering Gene Ontology (GO) biological processes as annotation set and *Danio rerio* as reference gene set. Only GO terms with FDR P < 0.05 were considered.

## Validation of RNA-Seq analyses

A further DNase treatment with PerfeCta® DNase I (RNase-free) kit (Quanta Biosciences, Germany) was performed on the previously extracted RNA samples. Then, approximately 450

**Table 1. Primers used for amplification of genes by qPCR.** Overview of full gene name, accession number and target sequences.

| Full name | Abbreviation | Accession no. | Primer (5′-3′) (F: Forward; R: Reverse) | Reference |
|---|---|---|---|---|
| Vitellogenin 1 | *VTG1* | EU073127 | F: GACAGTGTAGTGCAGATGAAG | [40] |
| | | | R: ATAGAGAGACAGCCCATCAC | |
| Vitellogenin 2 | *VTG2* | EU073128.1 | F: GATGCTCCCCTAAAGTTTGTG | [40] |
| | | | R: AGCGTCCAGAATCCAATGTC | |
| Beta-actin | *B-ACTIN* | DQ286836 | F: AGCCTTCCTTCCTGGGTATG | [40] |
| | | | R: GTTGGCGTACAGGTCCTTAC | |
| Heat shock protein 90 | HSP90 | AZBK01838994 | F: ACCATTGCCAAGTCAGGAAC | [41] |
| | | | R: ACTGCTCATCGTCATTGTGC | |
| Insulin like growth factor 1 | IGF1 | EU018410.1 | F: TTCCTCTTAGCTGGGCTTTG | [41] |
| | | | R: AGCACCAGAGAGAGGGTGTG | |
| Insulin like growth factor 2 | IGF2 | AZBK01717674 | F: ACAACGGATATGGAGGACCA | [41] |
| | | | R: GGAAGTGGGCATCTTTCTGA | |
| Elongation factor 1 alpha | EF1A | EU407824 | F: CTGAAGCCTGGTATGGTGGT | [40] |
| | | | R: CATGGTGCATTTCCACAGAC | |

ng of total RNA was retrotranscribed to cDNA using the qScript™ cDNA synthesis Kit (Quanta Biosciences, Germany). Six among upregulated and downregulated genes were selected (*VTG1*, *VTG2*, *ESR1*, *HSP90*, *IGF1* and *IGF2*) as well as two reference genes (Cytochrome c oxidase subunit I (*COX1*) and beta-actin (*β-actin*) that did not show significant changes in the mRNA-Seq analysis. Primers are listed in Table 1. Gene expression was performed with qPCR Biomark™ HD system (Fluidigm) in 96.96 IFC using four technical replicates. A pre-amplification step of cDNA was done following the Fluidigm protocol (PN 100–5875). Samples were diluted 1:5 before being loaded onto the arrays and the forward and reverse primers were loaded at a combined concentration of 100 μM. The arrays were run according to Fluidigm 96.96 IFC protocol (PN 100–9792) with a Tm of 60°C. The relative quantity of target gene transcripts was normalized to the geometric mean of the two reference genes. Gene expression was calculated according to the $2^{-\Delta\Delta CT}$ method with the control average as reference [39], then log fold change was calculated as follow:

$$\text{Log fold change} = \text{LOG}\frac{\overline{2^{-\Delta\Delta CT}week\ 9}}{\overline{2^{-\Delta\Delta CT}week\ 0}}$$

Correlation among fold changes of the genes of the two approaches (RNA-Seq and qPCR) was calculated with the ggpubr R package (https://CRAN.R-project.org/package=ggpubr).

## Results

### Phenotypic and RNA-Seq overview

The distribution of HSI across samples for the two sampling points is shown in Fig 1A. The average HSI increase was 1.6 fold, from 0.78 (± 0.10) in week 0 and to 1.25 (± 0.24) in week 9, i.e. increase of 80.59%, with a higher degree of variability in week 9. The rise in HSI was highly correlated to an increase in GSI (r = 0.93 P = 2.9e-07; Fig 1B) accompanying the transition of oocyte and ovarian developmental stage from previtellogenic in week 0 to late vitellogenic stage in week 9, as determined through histological analysis (Fig 2).

The sequencing and filtering produced approximately 25,600,000*2 paired reads for each sample (Table 2). Approximately 57% (±2.89) of these reads uniquely mapped against the eel

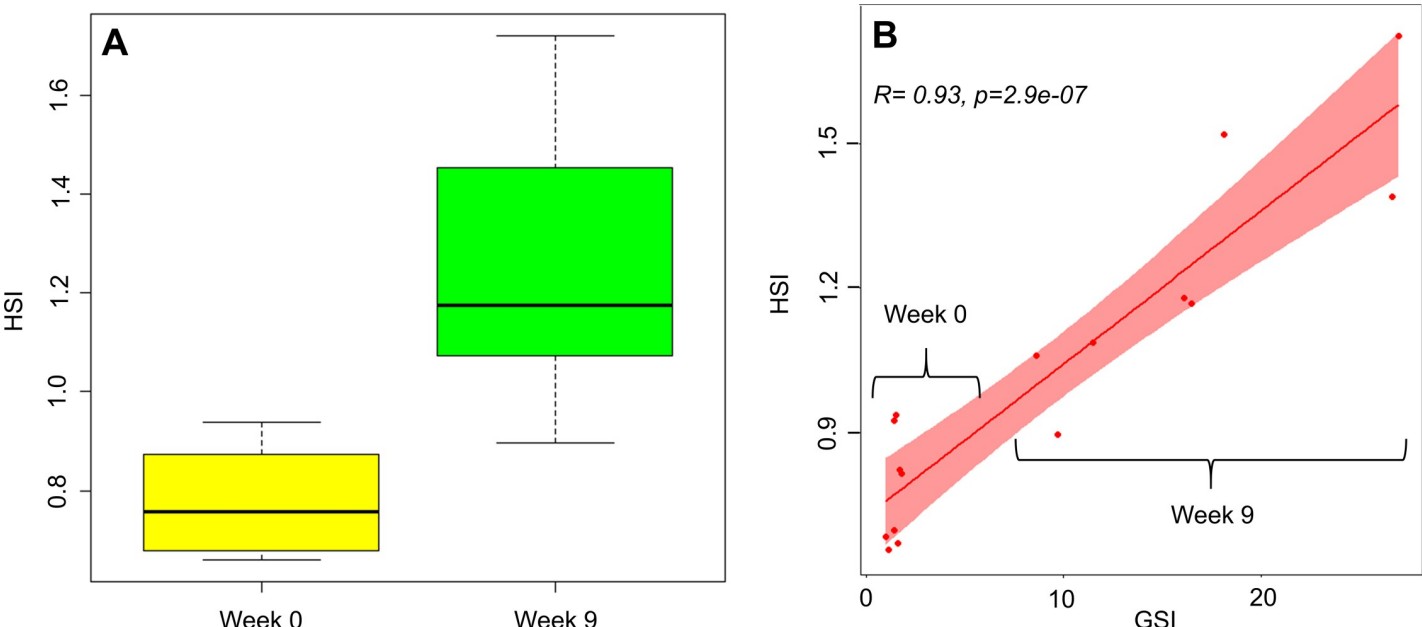

**Fig 1.** Statistical distribution of hepatosomatic index (HSI) of European eel before and during vitellogenesis (A) and correlation with gonadosomatic index (GSI) (B).

reference genome, and these results are in line with what was previously reported in a RNA-Seq analysis in ovaries of the same species and with the same reference genome [29]. The PCA plot based on the RNA-Seq analysis showed that samples from week 0 and 9 clustered into two separate groups (Fig 3). Samples from week 9 fluctuated more than those from week 0 in agreement with what has been observed with HSI. These variations may relate to differences in the individual response to the weekly administration of CPE to induce vitellogenesis.

## Differential expression

From 34,438 predicted annotated genes, a total of 1,328 upregulated and 1,490 downregulated genes were detected in the week 9 samples characterised by vitellogenic oocytes (Fig 4 and S1 Table). The most highly upregulated genes were vitellogenin genes (*VTG1*, and *VTG2*), with Log2fold change of 11.50 and 11.47, respectively.

Some of the genes differentially expressed have been previously detected in relation to European eel male reproductive development [26]. Common genes detected included serine palmitoyl transferase 2 (*SPTLC2*), elongation of very long chain fatty acids 1 (*ELOVL1*) and diacylglycerol O-acyltransferase 1 (*DGAT1*), which were downregulated in both studies. Conversely, mid1-interacting protein 1-like (*MID1IP1L*), acetyl-coenzyme A acetyltransferase 1-like (*ACAT1*), ATP-binding cassette sub-family D member 3 (*ABCD3*), ATP-synthases, glutamine synthetase, NADH dehydrogenase and hydroxysteroid dehydrogenase protein 2-like (*HSDL2*) and sulfotransferase 6B1-like gene (*SULT6B1*) were upregulated in both studies.

## Overrepresentation analyses

The PANTHER classification system was used for overrepresentation analyses of the differentially expressed transcripts. The classification according to biological processes was based on the function of the encoded protein in the context of a larger network of proteins that interact to accomplish a process at the cellular or organism level. Here, processes that came from upregulated and downregulated genes differed.

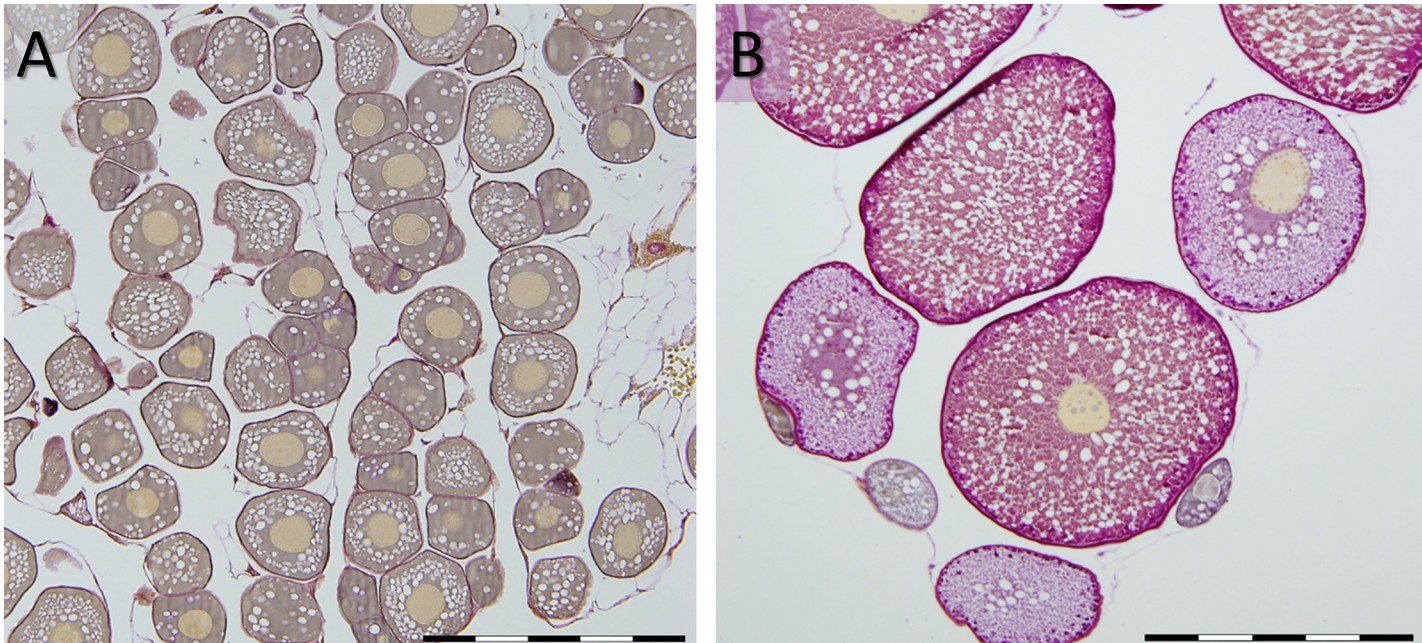

**Fig 2.** Micrographs of ovarian tissue of European eels sampled in Week 0 (A) and Week 9 (B). The week 0 tissue section shows previtellogenic oocytes in different stages of development, characterized by a yellow germinal vesicle and unstained lipid droplets in the cytoplasm, embedded among adipocytes. Week 9 tissue shows two cohorts of vitellogenic oocytes (early and late vitellogenesis) characterized by brownish-red yolk globules and lipid droplets in the expanded cytoplasm as well as remaining previtellogenic oocytes. Scale bars A: 250 μm, B: 500 μm.

The analysis conducted on the upregulated genes (Fig 5A, S2 Table) showed the presence of several processes related to biosynthesis (33 terms), such as lipids, cholesterol and organic compounds, isopentenyl diphosphate, aromatic compounds, nucleotides and nucleosides. Another process was related to mitochondrion activity reorganizations/energy generation and only present in upregulated genes (e.g. ATP biosynthesis and proton/electron transport) with

**Table 2. Summary of Illumina sequencing data and mapped reads for the samples.**

| Week | Sample name | Total Reads (n.) | Mapped reads (%) | Mapped reads unique (%) | Unmapped reads (%) |
|---|---|---|---|---|---|
| **Week 0** | Liver1 | 56,376,640 | 54.65 | 52.80 | 45.35 |
| | Liver2 | 51,679,828 | 54.46 | 52.51 | 45.54 |
| | Liver3 | 52,031,932 | 54.19 | 52.33 | 45.81 |
| | Liver4 | 57,653,658 | 53.66 | 51.80 | 46.34 |
| | Liver5 | 57,380,028 | 55.42 | 53.53 | 44.58 |
| | Liver6 | 39,401,684 | 51.90 | 50.22 | 48.10 |
| | Liver7 | 43,495,786 | 51.01 | 49.29 | 48.99 |
| | Liver8 | 43,493,164 | 53.53 | 51.81 | 46.47 |
| **Week 9** | Liver9 | 48,866,080 | 53.47 | 51.64 | 46.53 |
| | Liver10 | 53,694,222 | 51.86 | 49.79 | 48.14 |
| | Liver11 | 47,043,110 | 53.57 | 52.12 | 46.43 |
| | Liver12 | 54,471,766 | 56.87 | 55.07 | 43.13 |
| | Liver13 | 55,455,006 | 48.54 | 43.03 | 51.46 |
| | Liver14 | 57,780,420 | 55.61 | 53.97 | 44.39 |
| | Liver15 | 52,550,906 | 58.60 | 50.30 | 41.40 |
| | Liver16 | 47,857,560 | 58.96 | 56.85 | 41.04 |

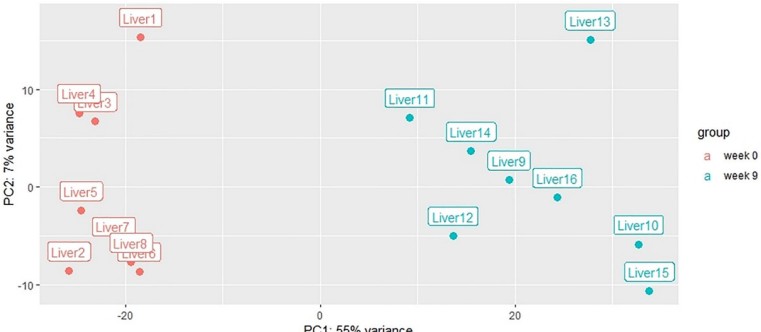

**Fig 3. PCA plot of the differential expression analysis for European eel liver from previtellogenic (sampled in week 0) and late vitellogenic females (sampled after 9 weeks of treatment).**

10 terms. Four biological processes were related to steroid production, including estrogen (steroid metabolism/biogenesis, response to estrogen stimulus) subcategories. This process included genes such as *VTG1* and *VTG2* (as mentioned above), Estrogen receptor 1 (*ESR1*), Lectin, Mannose Binding 1 (*LMAN1*) and Nuclear Protein 1, Transcriptional Regulator (*NUPR1*). Similarly, the GO biological processes related with localization were only present in the upregulated genes, including energy coupled proton transport and oxidative phosphorylation. Several of these upregulated genes were included in more than one biological process. Among those, Cytochrome C1 (*CYC1*) that was present in 65 biological processes, the hepatic fatty acid elongase-5 (*ELOVL5*) that was present in 58 biological processes, lysyl-tRNA synthetase (*KARS*) and Acyl-CoA synthetase short-chain family member 1 (*ACSS1*) that were included in 55 biological processes and Coenzyme A (CoA) synthase *(Coasy)* that was present in 46 biological processes.

Among the downregulated genes (Fig 5B, S3 Table), development and morphogenesis of organs (e.g. liver, hematopoietic, digestive system) and tissues and cells (e.g. axon, type B pancreatic cell, vascular tissues) were an important part of the GO biological processes (49 terms), not detected with the analysis performed in the upregulated genes. Moreover, a unique set of terms for the pathways of downregulated genes was the negative regulation of different general processes (18 terms) such as nitrogen compounds, transcription, biosynthesis of macromolecules and RNA. Other terms that were only overrepresented in the downregulated genes were immune system development and processes. Notch receptor 1a (*NOTCH1* or *NOTCH1A*) was present in 85 of the downregulated biological processes, Vascular endothelial growth factor (*VEGFA*) was present in 72 biological processes and Nuclear receptor corepressor 1 (*NCOR1*) was present in 51 processes.

## Gene expression validation

To validate the DEGs discovered through RNA-Seq analysis, the expression of eight genes were measured through qPCR. These included four genes that were upregulated in week 9 (*VTG1*, *VTG2*, *HSP90* and *ESR1*), two that were significantly downregulated (*IGF2* and *IGF1*) and two genes that were not significantly differentially expressed, which were used as reference genes (*EF1A* and *β-actin*). The selected genes showed similar expression pattern regardless of method. In fact, correlation among the differential expression pattern was high (R = 0.98, *p* = 0.00063; Fig 6), which validates the accuracy of the RNA-Seq analysis.

## Discussion

RNA-Seq analysis can provide an overview at the whole transcriptome level of genetic activities in several organisms under different environmental or physiological changes. This is

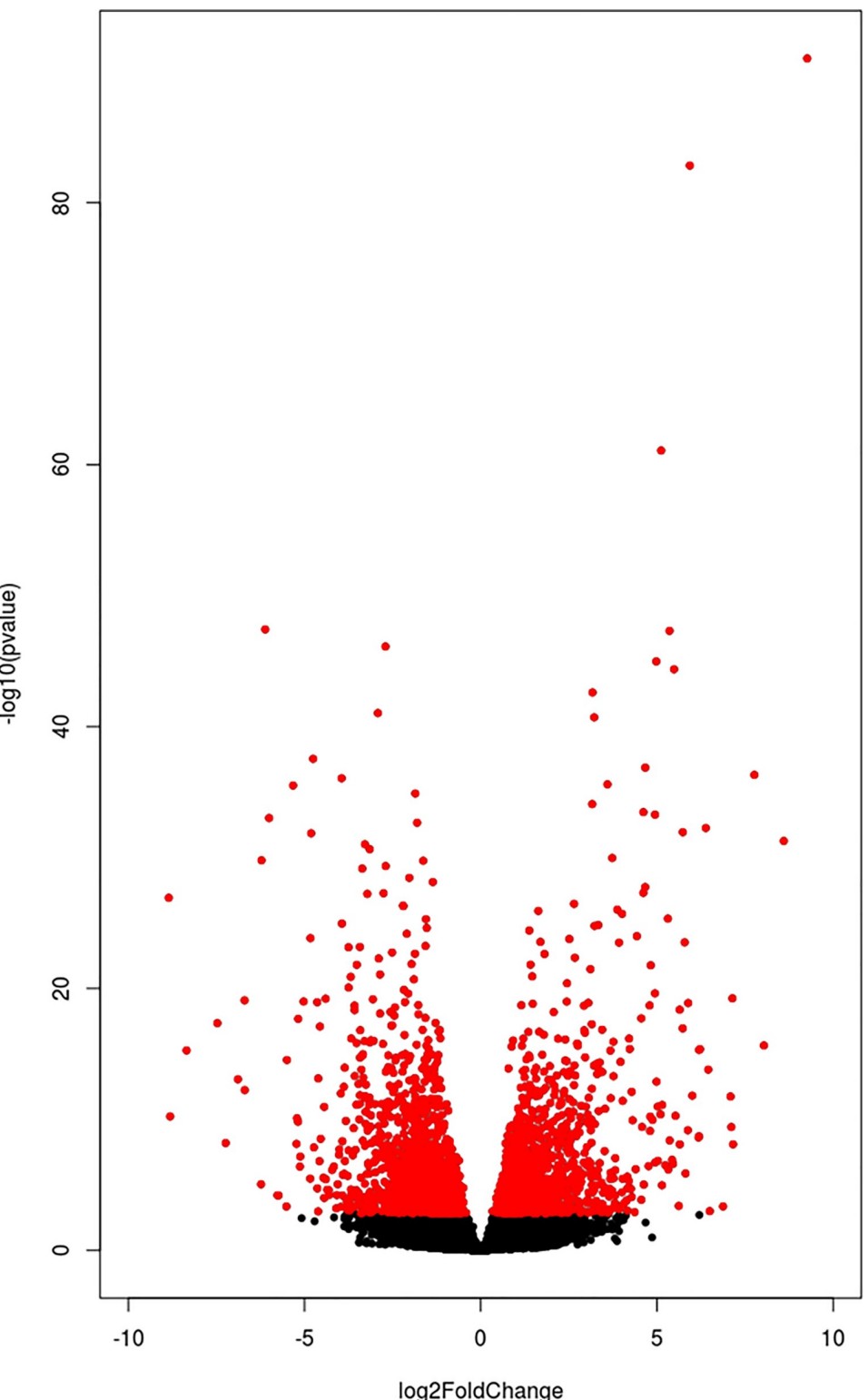

**Fig 4. Volcano plot of the differential expression analysis.** Each dot represents a gene. The x-axis reports the Log2fold change while the y-axis reports the–log10 of the adjusted P-value. All genes with adj. P < 0.01 were reported in red, with the left part (the negative part of the x-axis) representing the downregulated genes and the right part (the positive part of the x-axis) representing the upregulated genes.

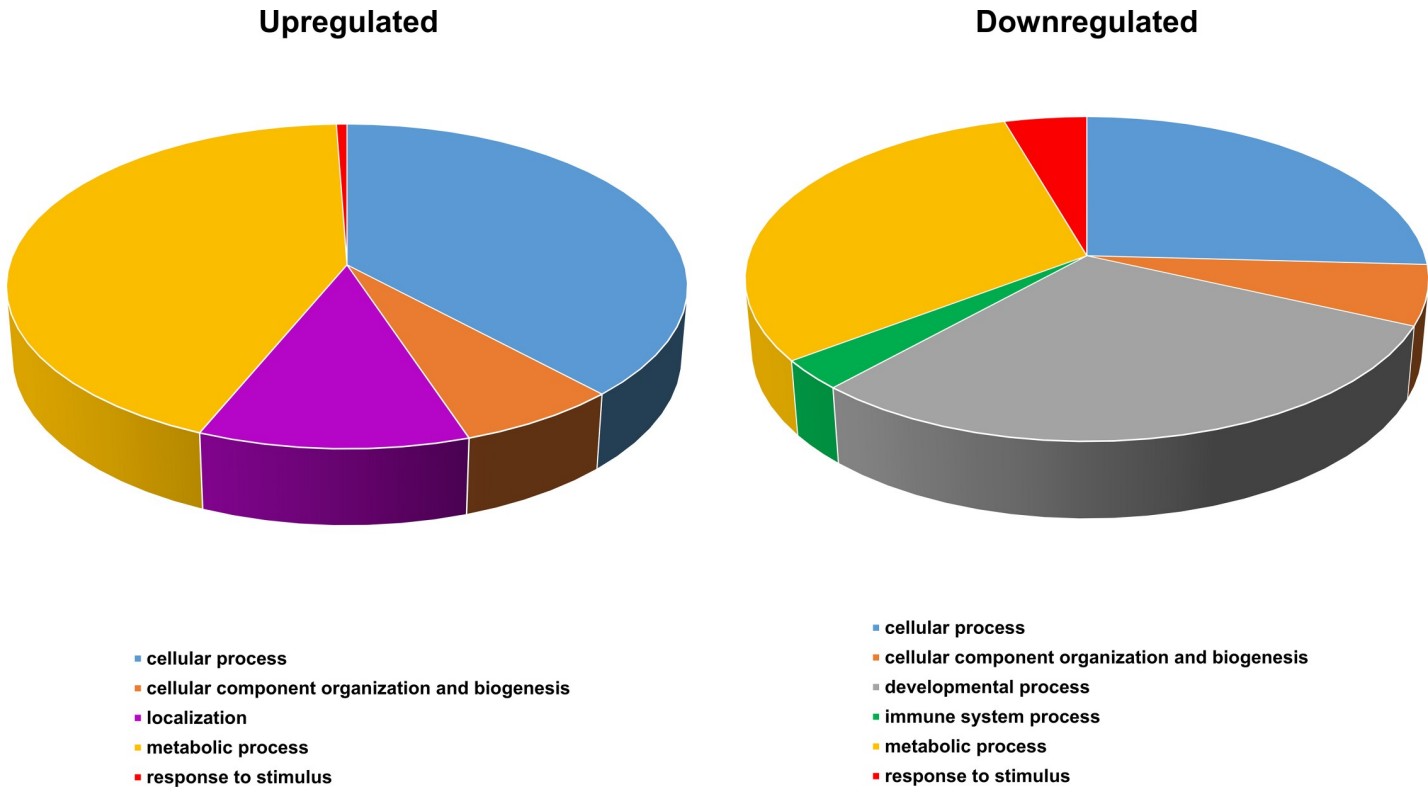

**Upregulated**

**Downregulated**

- cellular process
- cellular component organization and biogenesis
- localization
- metabolic process
- response to stimulus

- cellular process
- cellular component organization and biogenesis
- developmental process
- immune system process
- metabolic process
- response to stimulus

**Fig 5. The proportions of significantly enriched terms belonging to the biological processes, for upregulated and downregulated genes with FDR less than 5%.**

particularly useful to understand the molecular mechanisms of animals with an exeptional physiology such as the European eel. Together with the Japanese and American eels, they are among the few fish that do not feed during vitellogenesis [1]. This unusual aspect is interesting in the perspective of future management strategies of broodstock, as the efforts to complete the life cycle and rear these species in captivity are ever-increasing [17,18]. Moreover, eels represent a unique model to study vitellogenesis in such a distinctive condition. In female eels, the liver plays key roles as it is involved in the mobilization of reserves, including protein and lipids, to be processed and used for ovarian and follicular development including formation yolk globules in the developing oocytes and to supply energy necessary for maintenance and, in nature, for swimming to the Sargasso Sea and then spawn [42].

At a phenotypical level, the size of liver in teleosts tends to increase during vitellogenesis, as it has been shown in several fish species [43,44]. By contrast, the hepatosomatic index tends to decrease during lack of food/fasting, when vitellogenesis is not involved [45,46]. In nature, no significant differences in liver size have been detected among female yellow eels and eels in the prepubertal silvering stages of the European eel [8], in agreement with observations in Japanese eels [47]. Our data showed that, despite the absence of food during the trial, the dimension of liver increased during induced maturation, almost doubling its initial value. The increase in HSI during induced maturation is in agreement with previous studies in the European eel [48]. Moreover, our results showed a highly significant positive correlation between HSI and GSI.

When we looked at the differential expression analysis, a high number of genes were significantly different between week 0 and week 9. Some of the upregulated and downregulated DEGs were shared with the study performed on male eels [26]. These genes are mainly related

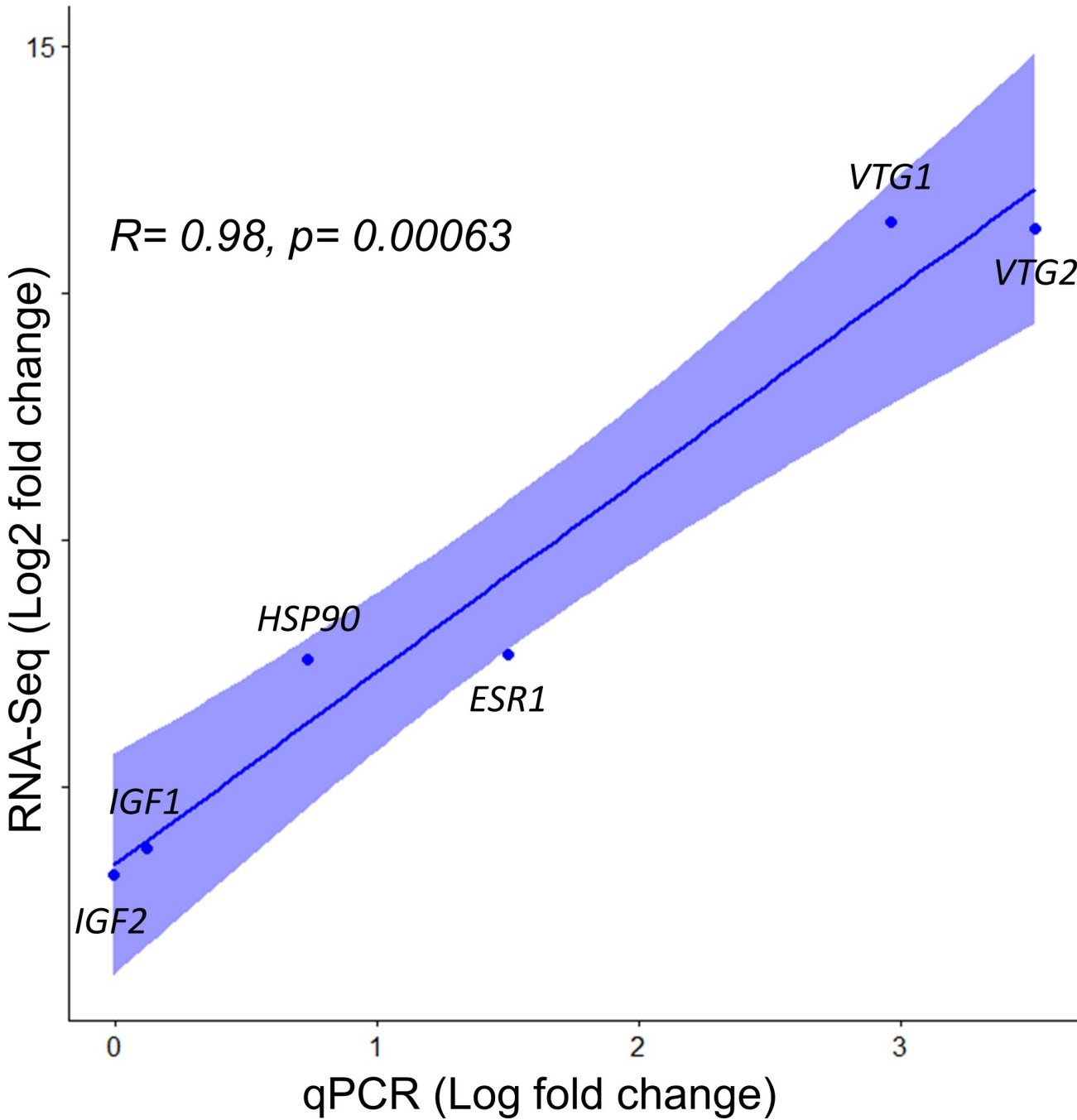

**Fig 6.** Correlation between the RNA-Seq (on the y-axis) and qPCR (on the x-axis) values for the same genes.

to energy management and biosynthesis of metabolites that may represent common requirements for energy and metabolite management and survival in this species. In agreement with [26], there is a distinction of the patterns of upregulated and downregulated genes, as upregulated genes were mainly involved in biosynthetic processes, localization and process related to reproduction, whereas downregulated were linked to developmental processes, organ maintenance and immune system.

## Reproduction: Response to estrogen

The brain-pituitary-gonadal-liver axis plays a crucial role in the reproductive processes of oviparous female teleosts. Here, the main function of liver is to produce vitellogenins, glycoli-poproteins that are deposited to the egg in the ovary through the bloodstream [23]. Increased transcription levels of vitellogenin genes in the liver have already been observed in several works on induced vitellogenesis of European eel [49,50]. Our results agree with these findings and provide clear evidence that after 9 weeks, the vitellogenin genes (*VTG1* and *VTG2*) have the highest increase in expression among all the other genes assayed.

The overrepresentation analysis indicated that, together with the vitellogenin genes, other genes grouped under the GO biological process response to estrogen. The group counted seven genes including *ESR1*, *LMAN1* and *NUPR1*. *ESR1* gene is well known, as it is the nuclear receptor that binds E2 to initiate the synthesis of vitellogenins [23,50]. The *NUPR1* gene encodes a chromatin-binding protein that is involved in the adaptation to stressor events. It has been investigated in the liver of zebrafish in relation to the response to chemical distruptors [51]. So far, no information on the involvement of this gene in fish reproduction is available. However, *NUPR1* plays a role in the temporal expression of the beta subunit of luteinizing hormone, during gonadotropin development in mice, as shown by knockout studies [52,53]. In teleosts, luteinizing hormone regulates oocyte maturation, where it leads to resumption of meiosis until metaphase II in oocytes. Moreover, luteinizing hormone acts on follicle cells to stimulate the synthesis of progesterone that will reactivate meiosis in the oocyte during final maturation [54–57]. Therefore, *NUPR1* may represent an interesting target for further investigation in teleost vitellogenesis.

As for *LMAN1*, an analysis among 2,523 transcripts and different vitellogenic stages in zebrafish, showed evidence that this gene is also upregulated in the liver of E2 treated males [58]. As in our study, Levi and colleagues reported the upregulation of *VTG1* and *ESR1* that were included in the estrogen receptor GO term, as well as other genes that were detected differentially expressed in our study. These genes are Reticulon *1 (RTN1)*, Prefoldin Subunit 6 (*PFDN6*) and Nitric Oxide Synthase Interacting Protein (*NOSIP*) genes that in our analysis were upregulated as well as Low Density Lipoprotein Receptor (*LDLR*) and Heterogeneous Nuclear Ribonucleoprotein H1 (*HNRNPH1*) that were downregulated.

In oviparous teleosts, zona pellucida (ZP) proteins are essential to the hardening process of the egg envelope after fertilization, providing physical protection against the environment. Depending on the species, ZP proteins can be produced in the liver, in the ovary or in both tissues [59]. The most recently available annotation of the European eel genome, used in our analysis, presented four ZP genes annotated and a ZP-like gene, but none of these was differentially expressed in the timeframe analyzed. These results were in agreement with a previous study in the European eel on two ZP genes analyzed over four time points in the liver (i.e. week 0, 4, 8 and 12), where no significant differential expression was detected [60]. Our results strengthen evidence that for European eel, the liver may not play a role in the production of ZP proteins during vitellogenesis.

## Metabolism

**Biosynthetic processes and energy.** Liver is responsible for many energy metabolic processes, including metabolic protein synthesis, degradation, and fatty acids biosynthesis [61]. Despite liver growth phenotypically followed the vitellogenic pattern, several DEGs and related biological processes were linked to response to fasting conditions. For example, *CYC1* was the upregulated gene that was present in the highest number of enriched biological processes, particularly nucleotide/nucleoside metabolic process and ATP metabolism. This gene seems to be

involved in adaptation to high temperatures in Redband trout [62] and is important for liver adaptation in prolonged fasting in humans [63]. Even if knowledge on the enzymatically pathways for eel is substantially unknown, European eel maintain the ability of lipid biosynthesis during fasting [64,65], and modification of fatty acids has been observed in maturing male eels, where de-novo synthesis of fatty acid has been suggested [66,67]. In this context, *ELOVL5* was upregulated in our work and was included in biological processes such as biosynthesis of several molecules. This gene has an important function in long-chain monounsaturated and polyunsaturated fatty acid synthesis. *ELOVL5* activity is regulated during development by diet, hormones and drugs. Furthermore, *ELOVL5* activity may affect hepatic glucose production during fasting [68]. *ELOVL5* was also highly expressed in the livers of laying hens and increased rapidly after sexual maturity [69].

*Coasy* plays a central role in cellular metabolism in agreement with our enrichment analysis, where it was enriched in biosynthetic processes of aromatic compounds, carbohydrate derivatives, nitrogen compounds, heterocyclic compounds and nucleotides. A downregulation of *Coasy* in zebrafish expressed mainly in the liver, muscles and head, affected neural and vascular development from early life stages [70]. Other genes that in our work were included in biosynthetic processes were *KARS*, differentially expressed in the muscle of trout when fasting [71] and *ACSS1*, important for maintaining normal body temperature during fasting and for energy homeostasis in mice [72].

**Morphology and development.** Food deprivation affects morphogenesis and development of several organs as these processes needs to be shut down to reallocate resources to produce energy for surviving and, in the case of eel, migration and vitellogenesis. In our analysis, downregulated genes were mainly included in development or morphogenesis of several organs including the liver, immune system, digestive system and sensory organs. The overrepresentation analysis may indicate a downregulation of genes that are important for the maintenance of several organs. It would be interesting to understand if the downregulation of genes related to these processes in the liver has an impact on the maintenance of other organs as suggested by the overrepresentation analysis (e.g. the gastrointestinal tract that in eel shrinks during fasting), but to our knowledge no information is yet available.

In teleosts as in other vertebrates including mammals, the liver acts as an endocrine gland involved in body growth. Under the stimulatory effect of pituitary growth hormone, the liver produces Insulin-like Growth Factors (*IGF*), which are released in the bloodstream and exert mitogenic activities on various tissues such as muscle and skeleton (for review: [73]). As shown in salmonids, fasting induces a decrease in liver *IGF1* transcripts and arrest in body growth, while refeeding leads to a rise in liver *IGF1* transcripts (e.g. [74]). The downregulation of the liver expression of *IGF1* and *IGF2* observed in our study highlights the inhibition of the "body growth function" of the liver during eel reproductive development.

In our work, *NOTCH1* was downregulated and was overrepresented in biological processes such as morphogenesis and development of several tissues and organs. This gene is involved in several processes, including animal organ development, chordate embryonic development and regulation of cell fate specification. In human, misregulation of this gene can cause liver cancer [75]. Activation of Notch signaling requires a direct contact between cells expressing Notch ligands and cells expressing Notch receptors [76]. It is interesting to observe that its receptor, jagged canonical Notch ligand 1b (*JAG1B*) was also downregulated in our analysis. In zebrafish, the inhibition of Jagged-mediated Notch signaling in the liver affects biliary tract development and generates multi-organ defects [77].

*VEGFA* that was downregulated in our study was mainly present in biological processes related to morphogenesis and development. In fact, this gene stimulates the proliferation of sinusoidal endothelial cells and hepatocytes during liver regeneration [78].

Liver was identified as one of the most important immune relevant organs in mammals and fish [61,79]. The presence of the immune system as GO term in the downregulated genes and not in the upregulated genes is in line with what was observed during induced reproductive development in male eel [26]. In our study, 53 genes related to the immune system processes or development were downregulated. One of the gene included is *NCOR1* that in our study was present also in other biological processes such as response to growth hormone, formation of blood and cellular components (hemopoiesis). Moreover, it plays a key role in reallocation of metabolic resources during fasting. In fact in mice, *NCOR1* regulates many enzymes involved in fatty acid oxidation, desaturation, and elongation [80–82] and is involved in regulating mitochondrial and peroxisomal fatty acid oxidation [83]. During fasting, the activity of this gene is suppressed by autophagy and allows the Peroxisome Proliferator Activated Receptor Alpha (*PPARα*) to induce lipid oxidation [84]. PPARα is in fact confirmed to be upregulated in our dataset. The interaction between NCOR1 and PPARα during fasting, extensively studied in mice, is still unknown in fish, but our analysis indicates that similar regulation may occur also in fish.

## Conclusion

Induced vitellogenesis of female European eel resulted in differential expression of genes involved in response to estrogen and reproduction, allowing gonad and oocyte development.

At the same time, since eels do not feed during this process, several differentially expressed genes were linked with the requirement of energy and metabolite reallocation, several of which were also differentially expressed during fasting in other species. Upregulated and downregulated genes were related to different biological processes. Upregulated genes were involved in biological processes related to the biosynthesis of several molecules, energy production, mobilization and molecule transport. In contrast, downregulated genes were involved in the development and morphogenesis and to the immune system, as an indication of biological processes that may have down-scaled activity during vitellogenesis in this complex and distinctive fish species. Altogether, the results provide for the first time a holistic view, at both a phenotypical and a high-throughput molecular level, of the changes that occur in the liver of the European eel during unusual vitellogenesis, where oocyte development is accompanied by ending the feeding regime. These findings represent the benchmark for further investigations of vitellogenesis processes.

## Supporting information

**S1 Table. List of significantly differentially expressed genes.** Aannotation number based on the assembly, log2fold_change, P-value, homologous uniprot name (homologus_uniprot) and homologous gene symbol (gene_symbol) is reported.
(XLSX)

**S2 Table. List of significantly enriched GO Biological processes from the overrepresentation analysis for the upregulated genes.** Biological processes, subcategories of the biological process, GO-TERM number, Enrichment score and Genes detected in our analysis are reported.
(XLSX)

**S3 Table. List of significantly enriched GO Biological processes from the overrepresentation analysis for the downregulated genes.** Biological processes, subcategories of the biological process, GO-TERM number, Enrichment score and Genes detected in our analysis are

reported.
(XLSX)

## Acknowledgments

We would like to thank Maria Krüger-Johnsen and Johanna S. Kottmann (Technical University of Denmark) and Anne Katrine Bruun Olesen (The Danish Aquaculture Organisation) for assistance with broodstock management and Dorte Meldrup (Technical University of Denmark) for assistance with laboratory work.

## Author Contributions

**Conceptualization:** Francesca Bertolini, Michelle Grace Pinto Jørgensen, Jonna Tomkiewicz.

**Data curation:** Francesca Bertolini.

**Formal analysis:** Francesca Bertolini.

**Funding acquisition:** Jonna Tomkiewicz.

**Investigation:** Francesca Bertolini, Michelle Grace Pinto Jørgensen.

**Methodology:** Francesca Bertolini, Jonna Tomkiewicz.

**Project administration:** Jonna Tomkiewicz.

**Resources:** Christiaan Henkel, Sylvie Dufour, Jonna Tomkiewicz.

**Supervision:** Jonna Tomkiewicz.

**Validation:** Michelle Grace Pinto Jørgensen.

**Visualization:** Francesca Bertolini.

**Writing – original draft:** Francesca Bertolini.

**Writing – review & editing:** Michelle Grace Pinto Jørgensen, Christiaan Henkel, Sylvie Dufour, Jonna Tomkiewicz.

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
