## [Decision Letter · Decision Letter 0]

29 Apr 2020

PONE-D-20-08055

Unravelling the changes during induced vitellogenesis in female European eel through RNA-Seq: what happens to the liver?

PLOS ONE

Dear Dr Bertolini,

Thank you for submitting your manuscript to PLOS ONE. After careful consideration, we feel that it has merit but does not fully meet PLOS ONE’s publication criteria as it currently stands. Therefore, we invite you to submit a revised version of the manuscript that addresses the points raised during the review process.

Your manuscript has been reviewed by two referees who are recognized experts in this field. Both reviewers have made several comments that you will have to consider in order to revise your manuscript accordingly.

We would appreciate receiving your revised manuscript by Jun 13 2020 11:59PM. To enhance the reproducibility of your results, we recommend that if applicable you deposit your laboratory protocols in protocols.io, where a protocol can be assigned its own identifier (DOI) such that it can be cited independently in the future. For instructions see: http://journals.plos.org/plosone/s/submission-guidelines#loc-laboratory-protocols

We look forward to receiving your revised manuscript.

Kind regards,

Hubert Vaudry

Academic Editor

PLOS ONE

Journal Requirements:

'This work was supported by the Innovation Fund Denmark [grant numbers 5184-00093B (EEL-HATCH) and 7076-00125B (ITS-EEL)].**'**

'The funders had no role in study design, data collection and analysis, decision to publish, or preparation of the manuscript.'

Additional Editor Comments (if provided):

Reviewers' comments:

Reviewer's Responses to Questions

**Comments to the Author**

1. Is the manuscript technically sound, and do the data support the conclusions?

Reviewer #1: Yes

Reviewer #2: Yes

2. Has the statistical analysis been performed appropriately and rigorously? 

Reviewer #1: Yes

Reviewer #2: Yes

3. Have the authors made all data underlying the findings in their manuscript fully available?

Reviewer #1: Yes

Reviewer #2: Yes

4. Is the manuscript presented in an intelligible fashion and written in standard English?

Reviewer #1: Yes

Reviewer #2: Yes

5. Review Comments to the Author

Reviewer #1: The manuscript describes a study on European eel vitellogenesis with a RNA sequencing approach, making a differential expression analysis of liver samples from yellow and silvering female eels.

In general, the experimental design and methodology seems good, the manuscript is well written, and the discussed ideas are relevant (at least considering the grade of hypothetical conclusions required for this type of results).

I included a few suggestions to be considered by the authors, and found some formal mistakes that should be corrected (especially references format!).

Introduction

L74. migration instead of migrating?

L76. 80´s or even before in some places

L87 Reference 18. For a more specific information on eel, you could use Mylonas et al., 2017 (http://dx.doi.org/10.1016/j.aquaculture.2016.04.021).

L101 (plus comments on L276 and L394). There is scarce information available on the enzymatic control of fatty acid biosynthesis in the eel. Baeza et al., 2014´s (http://dx.doi.org/10.1016/j.aquaculture.2014.03.045) and 2015´s studies (http://dx.doi.org/10.1016/j.cbpa.2014.11.022) played with the idea of a liver involved in the lipids mobilization from muscle to gonad during gonadal maturation, but also evidenced de novo synthesis of lipids in the liver (in especial palmitic acid and EPA). Can your results clarify something on the role of European eel fatty desaturases and elongases?, and maybe relating them with dietary necessities for this species?

L129, L142, L205. 20 ºC (separate number and unit)

L134. Why fish were pit-tagged?

L137. S.L.U.

L166. 150 bp

Results

L224. Maybe is better to talk of a 1.6 fold increase of HSI.

L228. Vitellogenic

L234. 57%

L249. ...in the pre-spawning samples.

L250. (VTG1 and VTG2)

L298. Aorta mention seems too specific. Vascular tissues??

Discussion

L330. ...and used for ovary development and egg yolk...

L356. Check the recent paper from Morini et al., 2020. (http://dx.doi.org/10.1017/S1751731119003355). In the European eel, up-regulation of VTG under estradiol treatment was observed in eel in vivo (Burzawa-Gérard and Dumas-Vidal, 1991) as well as in vitro by hepatocyte primary cultures (Lafont et al., 2016). Nevertheless, ovaries from E2-treated eels are not able to incorporate Vtg (Dufour et al., 1988), and CPE treatment is necessary to induce ovarian vitellogenesis (Mazzeo et al., 2014).

They found found that vtgr transcription already occurs during early PV of immature eel and is not further activated in vitellogenic oocytes. Can your (liver) data support this conclusion somehow?

L364. chemical disruptors?

L371. Morini et al. 2017 (http://dx.doi.org/10.1016/j.cbpa.2017.02.009) reported the expression of several progestin receptors in the liver of immature eels, and described the changes of the expression of these genes happening in the testes during hormonally-induced spermatogenesis. Does your data suggest differential expression of these genes in the liver induced by hormonal treatment in the females?.

References

In general, follow the journal´s format.

Full references (including volume and page numbering, doi and PMID) are required.

Write especies´s name in italics.

Avoid writing with capitals every word of the titles.

Delete “and” from the list of authors (limited to 6; rest: et al.).

Check abbreviated journal names.

L533. IUCN?

L550, 563. European

L572. It is still under review?. Add full reference or delete.

L662. Oncorhynchus

L681. Nutritional regulation of insulin-like growth factor-I mRNA expression in salmon tissues???

Reviewer #2: The experiments were well designed in the study. In my opinion, the findings based on genes included in biological processes of the study will provide new insights for vitellogenesis in the eel or reproductive studies to be considered in other fish species. However, authors should take attention some issues regarding manuscript content. Authors should make corrections by controlling the text carefully according to the journal's rules. Some mistakes, for example in line 122, the length of the fish (76 ± 136 3 cm) should be corrected. A representative figure taken from the gonadal sections (especially vitellogenic stage) pertained to week 9 should be inserted in the results with a control section (previtellogenic stage).

6. PLOS authors have the option to publish the peer review history of their article (what does this mean?). If published, this will include your full peer review and any attached files.

Reviewer #1: Yes: Juan F. Asturiano

Reviewer #2: No

---

## [Author Response · Author response to Decision Letter 0]

23 Jun 2020

Reviewer #1: The manuscript describes a study on European eel vitellogenesis with a RNA sequencing approach, making a differential expression analysis of liver samples from yellow and silvering female eels.

In general, the experimental design and methodology seems good, the manuscript is well written, and the discussed ideas are relevant (at least considering the grade of hypothetical conclusions required for this type of results).

I included a few suggestions to be considered by the authors, and found some formal mistakes that should be corrected (especially references format!).

A: We want to thank the reviewer for the time spent on reviewing the manuscript: please, find below the replies point by point.

Introduction

R1. L74. migration instead of migrating?

A: Agreed. We changed it.

R1: L76. 80´s or even before in some places

A: Agreed. We changed as follow “The European eel stock has drastically declined since the 1980’s or even before in some places”

R1: L87 Reference 18. For a more specific information on eel, you could use Mylonas et al., 2017 (http://dx.doi.org/10.1016/j.aquaculture.2016.04.021).

A: The manuscript suggested is mainly focused on male spermatogenesis and the main topic of discussion here is in female eel. In the light of that reference 17 from 2010 does not comprehend the recent 10 years significant advances, we refocused the references to focus specifically on European eel, including the following reviews:

- Tomkiewicz J, Politis SN, Sørensen SR, Butts IAE, Kottmann JS. European eel – an integrated approach to establish eel hatchery technology in Denmark. Eels Biol Monit Manag Cult Exploit Proc First Int Eel Sci Symp. 2019; 340–374. 

- Mordenti O, Casalini A, Parmeggiani A, Emmanuele P, Zaccaroni A. Captive breeding of the European eel: Italian review. Eels Biol Monit Manag Cult Exploit Proc First Int Eel Sci Symp. 2019; 317–339.

- Asturiano JF. Chapter 14 Improvements on the Reproductive Control of the European Eel. In: Yoshida M, Asturiano JF, editors. Reproduction in Aquatic Animals: From Basic Biology to Aquaculture Technology. Singapore: Springer Singapore; 2020. pp. 293–320. doi:10.1007/978-981-15-2290-1_15

R1. L101 (plus comments on L276 and L394). There is scarce information available on the enzymatic control of fatty acid biosynthesis in the eel. Baeza et al., 2014´s (http://dx.doi.org/10.1016/j.aquaculture.2014.03.045) and 2015´s studies (http://dx.doi.org/10.1016/j.cbpa.2014.11.022) played with the idea of a liver involved in the lipids mobilization from muscle to gonad during gonadal maturation, but also evidenced de novo synthesis of lipids in the liver (in especial palmitic acid and EPA). Can your results clarify something on the role of European eel fatty desaturases and elongases?, and maybe relating them with dietary necessities for this species?

A. Thank you for pointing these articles to us. We do believe that our work is not able to provide insight about dietary necessity for this species, but it could be a benchmark for future studies in this direction. Indeed, the upregulation of ELOVL5 and in general the presence of the enriched term linked with the biosynthesis of fatty acids goes in the direction of what is hypothesized in the mentioned papers, and we therefore included it in the discussion (L408-411).

R1. L129, L142, L205. 20 ºC (separate number and unit)

A. Agreed, We modified that.

R1. L134. Why fish were pit-tagged?

A. Fish were pit-tagged as individual, initial body weight was used as basis for dosing the CPE in the hormonal treatment. Hence, the individual females were recognized at the weekly injection by scanning of pittags.

R1. L137. S.L.U.

A. We corrected it.

R1. L166. 150 bp

A. We corrected it.

Results

R1. L224. Maybe is better to talk of a 1.6 fold increase of HSI.

A. Agreed. We added that, but we kept also the average measurement for clarity.

R1. L228. Vitellogenic

A. We corrected it.

R1. L234. 57%

A. We corrected it.

R1. L249. ...in the pre-spawning samples.

A. We added “in the week 9 samples characterized by vitellogenic oocytes”. Females would develop for several more weeks before spawning applying the present treatment scheme.

R1. L250. (VTG1 and VTG2)

A. We corrected it.

R1. L298. Aorta mention seems too specific. Vascular tissues??

A. We corrected it.

Discussion

R1. L330. ...and used for ovary development and egg yolk...

A. We added” for ovarian and follicular development including formation yolk globules in the developing oocytes”.

R1. L356. Check the recent paper from Morini et al., 2020. (http://dx.doi.org/10.1017/S1751731119003355). In the European eel, up-regulation of VTG under estradiol treatment was observed in eel in vivo (Burzawa-Gérard and Dumas-Vidal, 1991) as well as in vitro by hepatocyte primary cultures (Lafont et al., 2016). Nevertheless, ovaries from E2-treated eels are not able to incorporate Vtg (Dufour et al., 1988), and CPE treatment is necessary to induce ovarian vitellogenesis (Mazzeo et al., 2014).

They found found that vtgr transcription already occurs during early PV of immature eel and is not further activated in vitellogenic oocytes. Can your (liver) data support this conclusion somehow?

A. We thank you for this insight that elucidate the strong connection among liver and ovary. VTGR is not differentially expressed in the liver, but it is also true that VTGR is known to be expressed in the ovary. We believe that the liver is not the best target to look at VTGR behavior. However, we have another study on the way, where we investigate eel ovarian development and we already included and discussed it in this manuscript that helps to clarify VTGR production in the ovary.

R1. L364. chemical disruptors?

A. Agreed, it is more correct, therefore we changed it.

R1. L371. Morini et al. 2017 (http://dx.doi.org/10.1016/j.cbpa.2017.02.009) reported the expression of several progestin receptors in the liver of immature eels, and described the changes of the expression of these genes happening in the testes during hormonally-induced spermatogenesis. Does your data suggest differential expression of these genes in the liver induced by hormonal treatment in the females?.

A. We rechecked our list of genes and we did not detect progestins that were differentially expressed. After reading the manuscript, we decided not to include it in the discussion as our manuscript do not add insight nor confirm a clear trend, and we prefer to focus on the non-expression of ZP genes that is confirming a finding of Mazzeo I, Peñaranda DS, Gallego V, Hildahl J, Nourizadeh-Lillabadi R, Asturiano JF, et al. 2012. The list of all differentially expressed genes is hovewer available in Table S1 for further comparison. 

R1. References

In general, follow the journal´s format.

Full references (including volume and page numbering, doi and PMID) are required.

Write especies´s name in italics.

Avoid writing with capitals every word of the titles.

Delete “and” from the list of authors (limited to 6; rest: et al.).

Check abbreviated journal names.

A. Agreed. We download the correct reference code directly from the PlosOne website to be used in Mendeley and we manually checked.

R1. L533. IUCN?

A. This the name that International Union for Conservation of Nature has in the title 

R1. L550, 563. European

A. corrected

R1. L572. It is still under review?. Add full reference or delete.

A. We deleted the reference

R1. L662. Oncorhynchus

A. corrected

R1. L681. Nutritional regulation of insulin-like growth factor-I mRNA expression in salmon tissues???

A. Thank you for pointing this bug. We manually complete the reference

Reviewer #2: The experiments were well designed in the study. In my opinion, the findings based on genes included in biological processes of the study will provide new insights for vitellogenesis in the eel or reproductive studies to be considered in other fish species. However, authors should take attention some issues regarding manuscript content. Authors should make corrections by controlling the text carefully according to the journal's rules. Some mistakes, for example in line 122, the length of the fish (76 ± 136 3 cm) should be corrected. A representative figure taken from the gonadal sections (especially vitellogenic stage) pertained to week 9 should be inserted in the results with a control section (previtellogenic stage).

A. We thank the reviewer for the comments and suggestions. We controlled it and modified the it according with the journal guidelines and we added a representative histology picture illustrating the previtellogenic (week 0) and late vitellogenic stages (week 9) (Fig 2, line 230)

---

## [Decision Letter · Decision Letter 1]

8 Jul 2020

Unravelling the changes during induced vitellogenesis in female European eel through RNA-Seq: what happens to the liver?

PONE-D-20-08055R1

Dear Dr. Bertolini,

We’re pleased to inform you that your manuscript has been judged scientifically suitable for publication and will be formally accepted for publication once it meets all outstanding technical requirements.

Kind regards,

Hubert Vaudry

Academic Editor

PLOS ONE

Additional Editor Comments (optional):

Reviewers' comments:

Reviewer's Responses to Questions

**Comments to the Author**

1. If the authors have adequately addressed your comments raised in a previous round of review and you feel that this manuscript is now acceptable for publication, you may indicate that here to bypass the “Comments to the Author” section, enter your conflict of interest statement in the “Confidential to Editor” section, and submit your "Accept" recommendation.

Reviewer #1: All comments have been addressed

Reviewer #2: All comments have been addressed

2. Is the manuscript technically sound, and do the data support the conclusions?

Reviewer #1: Yes

Reviewer #2: Yes

3. Has the statistical analysis been performed appropriately and rigorously? 

Reviewer #1: Yes

Reviewer #2: Yes

4. Have the authors made all data underlying the findings in their manuscript fully available?

Reviewer #1: Yes

Reviewer #2: Yes

5. Is the manuscript presented in an intelligible fashion and written in standard English?

Reviewer #1: Yes

Reviewer #2: Yes

6. Review Comments to the Author

Reviewer #1: This second version reflects all the comments ans suggestions I previously did.

In my opinion, the manuscript is ready to be published.

Reviewer #2: All requested recommendations have been completed by authors. The revised manuscript can be accepted for publication.

7. PLOS authors have the option to publish the peer review history of their article (what does this mean?). If published, this will include your full peer review and any attached files.

Reviewer #1: **Yes: **Juan F. Asturiano

Reviewer #2: No

---

## [Editor Report · Acceptance letter]

24 Jul 2020

PONE-D-20-08055R1 

Unravelling the changes during induced vitellogenesis in female European eel through RNA-Seq: what happens to the liver? 

Dear Dr. Bertolini:

I'm pleased to inform you that your manuscript has been deemed suitable for publication in PLOS ONE. Congratulations! Your manuscript is now with our production department. 

Kind regards, 

on behalf of

Dr. Hubert Vaudry 

Academic Editor

PLOS ONE